# Factors for Predicting Noninvasive Ventilation Failure in Elderly Patients with Respiratory Failure

**DOI:** 10.3390/jcm9072116

**Published:** 2020-07-04

**Authors:** Min Jeong Park, Jae Hwa Cho, Youjin Chang, Jae Young Moon, Sunghoon Park, Tai Sun Park, Young Seok Lee

**Affiliations:** 1Department of Internal Medicine, Korea Medical Center, Guro Hospital, Seoul 08308, Korea; minjeong87@gmail.com; 2Department of Internal Medicine, Gangnam Severance Hospital, Yonsei University College of Medicine, Seoul 06273, Korea; jhchomd@gmail.com; 3Department of Pulmonary and Critical Care Medicine, Inje University Sanggye Paik Hospital, Seoul 01757, Korea; yjchang0110@gmail.com; 4Division of Pulmonology and Critical Care Medicine, Department of Internal Medicine, Chungnam National University Hospital, Chungnam National University College of Medicine, Daejeon 35015, Korea; diffable@hanmail.net; 5Department of Pulmonary, Allergy and Critical Care Medicine, Hallym University Sacred Heart Hospital, Anyang 14068, Korea; f2000tj@hallym.or.kr; 6Department of Internal Medicine, Hanyang University College of Medicine, Seoul 04763, Korea; integrin@hanmail.net; 7Division of Respiratory and Critical Care Medicine, Korea Medical Center, Guro Hospital, 148, Gurodong-ro, Guro-gu, Seoul 08308, Korea

**Keywords:** noninvasive ventilation, respiratory failure, aged, intubation, critical illness

## Abstract

Noninvasive ventilation (NIV) is useful when managing critically ill patients. However, it is not easy to apply to elderly patients, particularly those with pneumonia, due to the possibility of NIV failure and the increased mortality caused by delayed intubation. In this prospective observational study, we explored whether NIV was appropriate for elderly patients with pneumonia, defined factors that independently predicted NIV failure, and built an optimal model for prediction of such failure. We evaluated 78 patients with a median age of 77 years. A low PaCO_2_ level, a high heart rate, and the presence of pneumonia were statistically significant independent predictors of NIV failure. The predictive power for NIV failure of Model III (pneumonia, PaCO_2_ level, and heart rate) was better than that of Model I (pneumonia alone). Considering the improvement in parameters, patients with successful NIV exhibited significantly improved heart rates, arterial pH and PaCO_2_ levels, and patients with NIV failure exhibited a significantly improved PaCO_2_ level only. In conclusion, NIV is reasonable to apply to elderly patients with pneumonia, but should be done with caution. For the early identification of NIV failure, the heart rate and arterial blood gas parameters should be monitored within 2 h after NIV commencement.

## 1. Introduction

Noninvasive ventilation (NIV) refers to ventilatory support delivered via a nasal, full- face, or helmet mask, without bypassing the upper airway, which is different from tracheal intubation or tracheostomy [1,2,3,4]. NIV is attractive when treating acute respiratory failure (RF), as it avoids complications, such as ventilator-associated pneumonia associated with invasive mechanical ventilation [5,6]. Although a high-flow nasal cannula (HFNC) may substitute for NIV when treating critically ill patients with acute hypoxic RF, it does not supply any inspiratory pressure, unlike NIV; therefore, NIV is preferred for patients with acute hypercapnic RF [7,8,9,10,11].

The proportion of elderly patients in intensive care units (ICUs) is rapidly growing, because global populations are aging [12]. If elderly patients require mechanical ventilation, weaning may be more prolonged than for younger patients, because of respiratory muscle weakness, comorbidities, and altered mental status [13,14,15]. Prolonged weaning is associated with several complications that increase in-hospital mortality [16,17,18]. Thus, many clinicians prefer to use a noninvasive approach, such as NIV, in elderly patients with acute RF to avoid intubations. However, in elderly patients, particularly those with pneumonia, there is concern regarding the possibility of NIV failure and the increased mortality caused by delayed intubation [19,20]. Thus, in elderly patients with pneumonia, NIV may be delayed, because its utility in this setting remains controversial, and excessive secretion (e.g., due to pneumonia) is a contraindication for the procedure. We hypothesized that, if we could define the factors that predict NIV failure so that it can encourage early intubation, rather than the maintenance of NIV, NIV could be safely applied to elderly patients with acute RF due to pneumonia.

The aim of this study was to investigate whether NIV was appropriate for elderly patients with pneumonia, to define factors that independently predicted NIV failure, and to make an optimal model for prediction of NIV failure.

## 2. Materials and Methods

### 2.1. Study Overview

This was a prospective observational study conducted at 21 university-affiliated hospitals of South Korea from June 2017 to February 2018. Among ICU patients prescribed NIV in accordance with international guidelines, we selected a subset by applying inclusion and exclusion criteria. The NIV machines, interfaces, and circuits were selected by respiratory intensivists from among the equipment available; the intensivists also chose initial NIV settings and durations by reference to patient conditions and the international guidelines. Patients who deteriorated after NIV underwent tracheal intubation at the discretion of respiratory intensivists. The study was approved by the Institutional Review Boards (IRBs) of all participating hospitals and the Korea Medical Center IRB (approval no.: 2017GR1298). Written informed consent was obtained from all enrolled patients or their legal surrogates.

### 2.2. Patients

The inclusion criteria were age ≥ 65 years and NIV prescribed at admission or during the hospital stay because of acute hypoxic/hypercapnic RF or acute on chronic RF. The exclusion criteria were age <65 years; a lack of written informed consent; NIV use despite contraindications, such as myocardial infarction or post cardiac arrest; NIV use because of acute postoperative RF; NIV weaning; and NIV palliative therapy.

### 2.3. Definitions

Acute RF was divided into acute hypoxic and acute hypercapnic RF. Acute hypoxic RF was defined by a partial pressure of arterial oxygen (PaO_2_)/percentage of inspired oxygen (FiO_2_) ≤300 and a normal or low partial pressure of arterial carbon dioxide (PaCO_2_). Acute hypercapnic RF was defined as a PaCO_2_ >50 mmHg [21,22]. Acute on chronic RF was defined as acute deterioration of RF caused by various factors in patients on home ventilators because of chronic disease. Age and body mass index (BMI; weight divided by the height squared) were treated as continuous variables. Pneumonia was diagnosed based on typical symptoms (e.g., fever, cough), elevated levels of inflammatory markers, and pneumonic consolidation evident in chest X-rays. Cardiovascular disease was defined as a prior diagnosis of ischemic heart disease, heart failure, or hypertension. Cerebrovascular disease was defined as a previous diagnosis of cerebral infarction or hemorrhage. Other comorbidities were defined using the traditional indicators. ICU mortality was death between ICU admission and discharge. NIV success was defined as successful weaning and then discontinuation. For chronic NIV users, NIV success was defined as transfer to a general ward because of stable gas parameters (e.g., PaO_2_/FiO_2_ > 300 or PaCO_2_ < 50 mmHg) when on NIV and resolution of any acute exacerbation. NIV failure was defined as a transition to mechanical ventilation (via tracheal intubation or tracheostomy), a “hopeless” discharge on NIV, or death while on NIV.

### 2.4. Statistical Analysis

Descriptive statistics are presented as medians (the 25th to 75th percentiles) or as numbers (percentages). The Fisher’s exact test was used to analyze categorical data, and the Mann-Whitney *U*-test was used to compare continuous data. The Wilcoxon signed-rank test was employed to determine the effects of NIV on physiological and laboratory parameters. Logistic regression analyses were used to identify factors independently predicting NIV failure. Independent variables and those with *p*-values <0.1 in univariate analyses were included in multivariate analyses. The data are presented as adjusted odds ratios (ORs) with 95% confidence intervals (CIs). We sought a model that optimally predicted NIV failure. The discriminatory power of each tested model was assessed using the Harrell C-index and the area under the curve (AUC); the models were compared using a bootstrap method. A two-tailed *p*-value < 0.05 was taken to indicate significance. All statistical analyses were performed using SAS ver. 9.4 software (SAS Institute, Cary, NC, USA).

## 3. Results

### 3.1. Clinical Characteristics

During the study period, 200 patients received NIV therapy. Of these, 122 were excluded (refusal to consent, *n* = 36; age <65 years, *n* = 37; NIV weaning, *n* = 40; postoperative NIV, *n* = 3; NIV after myocardial infarction, *n* = 1; NIV after cardiac arrest, *n* = 3; and palliative NIV, *n* = 2); thus, 78 patients were included, of whom 46 patients experienced NIV success and 32 patients experienced NIV failure (Figure 1).

The clinical characteristics are listed in Table 1. The median age was 77 years and 43 (55.1%) were male. The BMI value and sequential organ failure assessment (SOFA) score at NIV commencement were 20 and 4, respectively. Most patients (76.9%) had been diagnosed with cardiovascular disease; one-third had been diagnosed with diabetes mellitus and chronic kidney disease. In all, 32 patients (41%) had chronic obstructive pulmonary disease (COPD) and 20 (25.6%) had no underlying lung disease. Patients who experienced NIV success versus failure did not differ demographically. The median NIV duration and ICU stay were 3 and 8 days. Patients for whom NIV was successful exhibited longer-duration NIV use and a shorter ICU stay (because they did not require intubation) than patients for whom NIV failed. Seventeen patients (21.8%) died during ICU admission.

### 3.2. Comparison of Treatment of Patients for Whom NIV Succeeded and Failed

The NIV treatments of the two groups are compared in Table 2. Most NIV patients had hypercapnic RF and more patients with hypercapnic RF experienced NIV success than failure. The median PaO_2_/FiO_2_ ratio and PaCO_2_ values of these patients were 211 and 65 mmHg, respectively. Considering heart and respiratory rates, the PaO_2_/FiO_2_ ratio, and PaCO_2_, patients for whom NIV was successful had milder disease than those for whom NIV failed. Most patients were initially prescribed NIV in an ICU; patients prescribed NIV in general wards were transferred to ICUs. Most patients used oronasal masks. Typically, the initial NIV setting was pressure support mode; the initial IPAP and EPAP settings were 15 and 5 cm H_2_O. Eleven patients required sedatives during NIV; NIV was delivered for 15 h/day. Almost 15% of patients experienced skin erythema and large leakages, but the two groups did not differ significantly.

### 3.3. Causes and Outcomes in Patients with NIV Failure

The causes and outcomes of NIV failure are shown in Table 3. Among the 32 patients, 24 (75%) experienced NIV failure, because of an aggravated clinical condition. Five patients complained of discomfort due to NIV commencement; NIV was discontinued in these patients. Most patients with NIV failure were intubated and received mechanical ventilation. Nine patients died during NIV.

### 3.4. Factors Analyzed for Effects on NIV Success/Failure

Patients with pneumonia at admission were significantly more likely to experience NIV failure, compared to those without pneumonia (Appendix A; *p =* 0.020). We used logistic regression to identify factors associated with failure. Univariate analyses showed that the heart and respiratory rates, the PaO_2_/FiO_2_ ratio, PaCO_2_, use of the assist control mode, and pneumonia were significantly associated with NIV failure. Multivariate logistic regression (using backward elimination) revealed that a low PaCO_2_ (OR, 0.95; 95% CI, 0.910–0.982; *p* = 0.004), a high heart rate (OR, 1.05; 95% CI, 1.013–1.080; *p* = 0.006), and pneumonia (OR, 3.32; 95% CI, 1.026–10.719; *p* = 0.045) were significant independent predictors of NIV failure (Table 4).

### 3.5. Prognostic Utilities of Models Using Risk Factors for NIV Failure

The Harrell C-index was used to compare different combinations of factors, and pneumonia status, in terms of NIV failure prediction. Model I included only pneumonia; Model II included pneumonia and heart rate; and Model III included pneumonia, PaCO_2_ level, and heart rate. Model I predicted NIV failure poorly (C-index AUC: 0.639; 95% CI: 0.534–0.744). Models II and III were slightly better (Model II: 0.744; Model III: 0.816). The differences in the AUCs of Models II and III (compared to that of Model I) were significant (Model II: *p* = 0.025; Model III: *p* = 0.001) (Table 5).

Therefore, not only pneumonia status, but also a vital sign (heart rate) and arterial blood gas status (PaCO_2_) between 30 min and 2 h after NIV commencement, should be considered when predicting NIV failure.

### 3.6. Changes in Physiological Parameters after NIV Commencement

We collected arterial blood gas and vital sign data before NIV and between 30 min and 2 h after NIV commencement (Table 6). Patients for whom NIV was successful exhibited significantly improved heart rates and arterial pH and PaCO_2_ levels after NIV application. Patients for whom NIV failed evidenced a significantly improved PaCO_2_ level only. Therefore, patients without marked clinical improvement in both vital signs and arterial blood gas values 30–120 min after NIV commencement are at risk of NIV failure.

## 4. Discussion

We explored whether NIV was appropriate for elderly patients with pneumonia, defined factors that independently predicted NIV failure, and built an optimal model for prediction of such failure. Pneumonia was associated with NIV failure, but the presence of pneumonia alone did not predict such failure well (C-index AUC: 0.639; 95% CI: 0.534–0.744). Therefore, NIV is not contraindicated by pneumonia alone. Improvement of heart rate and PaCO_2_ level measured within 2 h after NIV commencement should be considered when determining if NIV should be continued or if intubation is preferable.

Our study makes several important contributions to the literature. When elderly patients are admitted to ICUs, intensivists find it difficult to decide whether aggressive treatment is appropriate, because advanced age is associated with poorer outcomes after mechanical ventilation, and survival does not always ensure a satisfactory quality of life [12,13,14,15,19,20,23,24]. Some intensivists prefer to treat elderly patients as noninvasively as possible, often choosing NIV for those with acute RF. However, some may hesitate to use NIV in elderly patients with pneumonia on the basis of a vague concern that NIV failure is more common in elderly patients than in younger patients [25,26]. NIV failure is associated with poor mask-fitting, claustrophobia, excessive secretions, intolerance, agitation, and patient/ventilator asynchrony, most of which may be associated with poor respiratory muscle power [26,27]. A previous study also showed that weak respiratory muscle power was associated with NIV failure [27]. Generally, older patients have weaker respiratory muscles than younger patients. Thus, concerns that NIV failure is more common in elderly patients than in younger ones may be partially right. However, some elderly patients remain active and have powerful respiratory muscles. Therefore, NIV should not be avoided simply because of patient age. In our study, age subgrouping (65–74, 75–84, and ≥85 years) did not affect the NIV failure rate (Appendix A; *p* = 0.535).

Early identification of NIV failure is important; delayed identification increases mortality. In a previous study, the timing of NIV failure was categorized as follows: (1) immediate failure (within minutes to < 1 h), (2) early failure (1–48 h), and (3) late failure (after 48 h) [26]. The majority (~83%) of NIV failures occur between the immediate and early stages [26]. When evaluating possible NIV failure, it is necessary to consider the overall clinical condition, but we could hesitate to perform intubation if NIV failure is not definitive. We emphasize that early variation in vital signs and arterial blood gas data measured 30–120 min after NIV commencement should be used to predict NIV possible failure, triggering intubation, thus preventing mortality attributable to delayed intubation.

We found that heart rate and PaCO_2_ level independently predicted NIV failure. Although the PaO_2_/FiO_2_ ratio has been associated with NIV failure in patients with hypoxic respiratory failure, the arterial pH and PaCO_2_ values of patients with hypercapnic respiratory failure were more closely associated with NIV failure than the PaO_2_/FiO_2_ ratio [26]. In our study, 82.1% of patients were prescribed NIV, because of hypercapnic respiratory failure. Therefore, the PaCO_2_ value more meaningfully predicted NIV failure than the PaO_2_/FiO_2_ ratio.

In this study, a low PaCO_2_ level after NIV commencement was associated with NIV failure. A low PaCO_2_ level was associated with severe disease and agitation, because the increased tidal volume and tachypnea caused by labored breathing led to reduction of the PaCO_2_ level [28,29]. A PaCO_2_ level after NIV commencement was related to a PaCO_2_ level before NIV commencement. Therefore, patients with a low PaCO_2_ level after NIV commencement initially had more severe disease, compared to patients with a high PaCO_2_ level after NIV commencement. Disease severity was associated with NIV failure, so NIV failure was more likely in patients with a low PaCO_2_ level than in patients with a high PaCO_2_ level. A more important finding related to the evaluation of NIV failure was the variation in PaCO_2_ level after NIV commencement. As shown in Table 6, variation in the PaCO_2_ level in the NIV failure group was associated with a less-improved arterial pH than that of the NIV success group. In other words, although the absolute median variation was similar in both groups, PaCO_2_ improvement was less clinically effective in the NIV failure group than in the NIV success group. This finding is probably because the number of patients with hypoxic respiratory failure was higher in NIV failure group than in NIV success group. However, a PaCO_2_ level reflects the clinical condition. Consequently, when predicting NIV failure, we should consider whether a PaCO_2_ level has improved properly to the point where pH reaches to the normal range. The results of a subgroup analysis of patients with only hypercapnic RF were similar to the findings in the present study (Appendix A). The heart rate was also a significant independent predictor of NIV failure, because a high heart rate was associated with disease severity and agitation. Previous studies have reported that respiratory rate variation is associated with NIV failure [26]. However, in practice, respiratory rate measurement is less accurate than heart rate measurement, because the former is not automatic. Both the respiratory and heart rates are affected by clinical condition. Therefore, heart rate variation is preferable to respiratory rate variation when predicting NIV failure.

We also explored the utility of NIV in patients with pneumonia. This has been controversial; prior studies have reported conflicting results, probably because of differences in clinical characteristics (pneumonia severity, age, or comorbidities) [30,31,32,33,34,35,36,37]. In patients with pneumonia, the median PaO_2_/FiO_2_ ratio and the median SOFA score at NIV commencement were 155 and 5. All pneumonia patients of the present study suffered from sepsis. In addition, 55% of patients had underlying lung parenchymal disease (COPD, asthma, interstitial lung disease, or bronchiectasis). Therefore, pneumonia was severe. We found that the C index of pneumonia alone was 0.639; thus, pneumonia was only weakly associated with NIV failure. Therefore, NIV is reasonable to apply in elderly patients with pneumonia, if we could carefully observe patients after NIV commencement.

Our study had certain limitations. First, the patient number was relatively small (*n* = 78). However, the number was sufficient to reveal the factors associated with NIV failure in elderly patients with acute RF, because a post hoc power analysis revealed that the power of each variable was appropriate. Second, we did not include clinical data, such as chest X-ray or laboratory findings, when determining disease severity in patients with pneumonia. However, given the SOFA scores and PaO_2_/FiO_2_ ratios, most pneumonia patients suffered from sepsis. Third, we did not set detailed indications for weaning or intubation, relying rather on respiratory intensivist discretion. However, all intensivists had extensive experience with NIV, and most decisions were based on international guidelines.

## 5. Conclusions

NIV is reasonable to apply in elderly patients with pneumonia, if we could carefully observe patients after NIV commencement. For the early identification of NIV failure triggering immediate intubation, the heart rate and arterial blood gas parameters should be closely monitored between 30 min and 2 h after NIV commencement. Additional studies with larger patient cohorts are warranted to validate our findings.

## Figures and Tables

**Figure 1 jcm-09-02116-f001:**
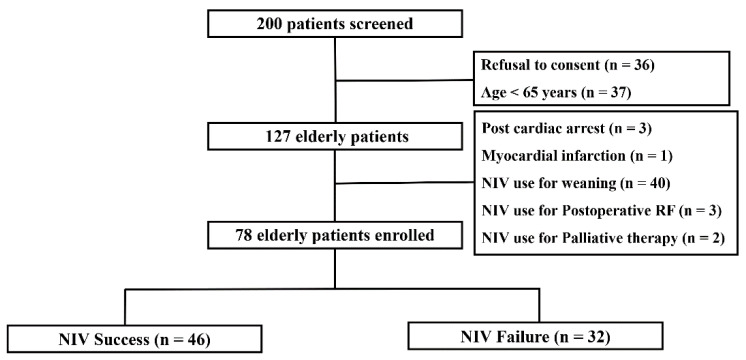
Flow chart of enrolled patients. NIV, noninvasive ventilation; RF, respiratory failure.

**Table 1 jcm-09-02116-t001:** Clinical characteristics of patients in this study.

Variables	Total	Noninvasive Ventilation		*p*-Value
	*n* = 78	Success (*n* = 46)	Failure (*n* = 32)	
Age, years *	77 (72–82)	78 (73–82)	76 (70–82)	0.542
Male	43 (55.1)	27 (58.7)	16 (50)	0.494
Body mass index, kg/m^2^ *	20 (17–24)	20 (17–24)	21 (17–25)	0.699
SOFA score at the start of NIV *	4 (2–5)	3 (2–5)	4 (2–6)	0.216
Comorbidity				
Cardiovascular disease	60 (76.9)	26 (56.5)	17 (53.1)	0.820
Diabetes mellitus	21 (26.9)	11 (23.9)	10 (31.3)	0.605
Chronic kidney disease	14 (17.9)	8 (17.4)	6 (18.8)	1.000
Liver cirrhosis	3 (3.8)	1 (2.2)	2 (6.3)	0.565
Cerebrovascular disease	9 (11.5)	5 (10.9)	4 (12.5)	1.000
Cancer	7 (9)	3 (6.5)	4 (12.5)	0.436
Immunosuppression	6 (7.7)	1 (2.2)	5 (15.6)	0.040
Underlying lung diseases				
No underlying lung disease	20 (25.6)	9 (19.6)	11 (34.4)	0.189
COPD	32 (41)	21 (45.7)	11 (34.4)	0.357
Bronchial asthma	2 (2.6)	1 (2.2)	1 (3.1)	1.000
Interstitial lung disease	5 (6.4)	2 (4.3)	3 (9.4)	0.396
Bronchiectasis	3 (3.8)	3 (6.5)	0 (0)	0.265
TB destroyed lung disease	8 (10.3)	4 (8.7)	4 (12.5)	0.710
Neuromuscular disease	1 (1.3)	0 (0)	1 (3.1)	0.410
Obesity disease	4 (5.1)	4 (8.7)	0 (0)	0.140
Scoliosis	1 (1.3)	1 (2.2)	0 (0)	1.000
Others chest wall disease	2 (2.6)	1 (2.2)	1 (3.1)	1.000
Total duration of NIV use, days *	3 (2–8)	4 (2–7)	2 (1–5)	0.042
Duration of ICU stay, days *	8 (5–20)	7 (4–11)	21 (7–40)	<0.001
ICU mortality	17 (21.8)	0 (0)	17 (53.1)	<0.001

Abbreviations: SOFA, sequential organ failure assessment; NIV, non-invasive ventilation; COPD, chronic obstructive pulmonary disease; TB, tuberculosis; ICU, intensive care unit. * Data are presented as median (25th percentile–75th percentile). Other variables are presented as number (percent).

**Table 2 jcm-09-02116-t002:** Comparison of treatment of patients for whom NIV succeeded and failed.

Variables	Total	Noninvasive Ventilation		*p*-Value
	*n* = 78	Success (*n* = 46)	Failure (*n* = 32)	
Cause of the NIV application				
Hypercapnic RF	54 (69.2)	35 (76.1)	19 (59.4)	0.139
Hypoxic RF	13 (16.7)	3 (6.5)	10 (31.3)	0.006
Acute on chronic RF	11 (14.1)	8 (17.4)	3 (9.4)	0.510
Physiologic parameters on the application of NIV *				
Systolic BP (mmHg)	125 (108–144)	123 (106–142)	126 (112–150)	0.318
Heart rate (beats/min)	98 (83–110)	92 (82–106)	107 (85–117)	0.051
Respiratory rate (breaths/min)	24 (20–29)	24 (20–28)	27 (23–32)	0.030
Arterial pH	7.34 (7.28–7.39)	7.34 (7.29–7.38)	7.34 (7.27–7.43)	0.830
PaO_2_/FiO_2_ ratio	211 (139–264)	214 (142–270)	199 (100–259)	0.152
PaCO_2_ (mmHg)	65 (50–80)	69 (59–83)	53 (43–68)	0.003
Location of initial NIV application				0.131
Intensive Care Unit	70 (89.7)	39 (84.8)	31 (96.9)	
General ward	8 (10.3)	7 (15.2)	1 (3.1)	
Interface type				
Nasal type	3 (3.8)	2 (4.3)	1 (3.1)	1.000
Oronasal type	68 (87.2)	41 (89.1)	27 (84.4)	0.732
Total facial type	1 (1.3)	0 (0)	1 (3.1)	0.410
Helmet type	6 (7.7)	3 (6.5)	3 (9.4)	0.685
Initial mode				0.119
Assist control mode	21 (26.9)	9 (19.6)	12 (37.5)	
Pressure support mode	57 (73.1)	37 (80.4)	20 (62.5)	
Initial setting *				
IPAP (cmH_2_O)	15 (12–18)	14 (12–17)	15 (13–20)	0.199
EPAP (cmH_2_O)	5 (4–6)	5 (4–5)	5 (4–6)	0.129
Tidal volume (mL)	400 (309–514)	390 (303–505)	410 (334–569)	0.308
Sedative use	11 (14.1)	5 (10.9)	6 (18.8)	0.344
Complications during NIV				
Skin erythema	12 (15.4)	8 (17.4)	4 (12.5)	0.752
Abdominal distension	4 (6.4)	3 (6.5)	2 (6.3)	1.000
Dry mouth	3 (3.8)	2 (4.3)	1 (3.1)	1.000
Aspiration	2 (2.6)	2 (4.3)	0 (0)	0.510
Claustrophobia	1 (1.3)	1 (2.2)	0 (0)	1.000
Nasal congestion	1 (1.3)	0 (0)	1 (3.1)	0.410
Large leaks	11 (14.1)	5 (10.9)	6 (18.8)	0.344
Duration of NIV (hours/day) *	15 (7–22)	16 (10–21)	12 (3–24)	0.449

Abbreviations: NIV, non-invasive ventilation; RF, respiratory failure; BP, blood pressure; PaO_2_, arterial partial pressure of oxygen; FiO_2_, fraction of inspired oxygen; PaCO_2_, arterial partial pressure of carbon dioxide; IPAP, inspiratory positive airway pressure; EPAP, expiratory positive airway pressure. * Data are presented as median (25th percentile–75th percentile). Other variables are presented as number (percent).

**Table 3 jcm-09-02116-t003:** Causes and Outcomes in patients with NIV failure (*n* = 32).

Variables	Number (%)
Cause of NIV failure	
Aggravated clinical conditions	24 (75)
Agitation	1 (3.1)
Aspiration	2 (6.3)
Patients discomfort	5 (15.6)
Outcomes after NIV failure	
Intubation & Mechanical ventilation	22 (68.8)
Hopeless discharge with NIV	1 (3.1)
Death during NIV treatment	9 (28.1)

Abbreviations: NIV, non-invasive ventilation.

**Table 4 jcm-09-02116-t004:** Factors analyzed for effects on NIV success/failure.

Variables	Odds Ratios	95% CI	*p*-Value
**Univariate Analysis**			
Age (years)	0.99	0.929–1.055	0.753
Male	0.70	0.284–1.745	0.448
Body mass index (kg/m^2^)	0.99	0.912–1.072	0.787
SOFA score at the start of NIV	1.17	0.955–1.433	0.130
Systolic blood pressure (mmHg)	1.02	0.995–1.036	0.136
Heart rate (beats/min) after NIV	1.04	1.009–1.066	0.010
Respiratory rate (breaths/min) after NIV	1.09	1.001–1.187	0.047
PaO_2_/FiO_2_ ratio after NIV	0.99	0.988–1.001	0.074
PaCO_2_ (mmHg) after NIV	0.95	0.916–0.980	0.002
The number of comorbidities	1.29	0.849–1.966	0.232
Assist control mode	2.47	0.888–6.849	0.083
IPAP	1.06	0.943–1.185	0.344
EPAP	1.23	0.904–1.667	0.189
Sedative use	1.89	0.524–6.837	0.330
Large leak	1.89	0.524–6.837	0.330
NIV application time during a day	0.97	0.920–1.031	0.360
The presence of pneumonia	3.69	1.314–10.389	0.013
**Multivariate Analysis**			
PaCO_2_ (mmHg) after NIV	0.95	0.910–0.982	0.004
Heart rate (beats/min) after NIV	1.05	1.013–1.080	0.006
The presence of pneumonia	3.32	1.026–10.719	0.045

Abbreviations: CI, confidence interval; SOFA, sequential organ failure assessment; NIV, non-invasive ventilation; PaO_2_, arterial partial pressure of oxygen; FiO_2_, fraction of inspired oxygen; PaCO_2_, arterial partial pressure of carbon dioxide; IPAP, inspiratory positive airway pressure; EPAP, expiratory positive airway pressure. Multivariate logistic regression analysis that used backward elimination was performed to predict noninvasive ventilation failure after adjusting for six variables (Heart rate (beats/min) after NIV, Respiratory rate (breaths/min) after NIV, PaCO_2_ (mmHg) after NIV, PaO_2_/FiO_2_ ratio after NIV, Assist control mode and the presence of pneumonia).

**Table 5 jcm-09-02116-t005:** Prognostic utilities of models using risk factors for noninvasive ventilation failure.

Models	C-Index	95% CI	*p*-Value *	*p*-Value ^†^
I	0.639	0.534–0.744	Reference	
II	0.744	0.630–0.859	0.025	Reference
III	0.816	0.715–0.917	0.001	0.052

Model I included only the presence of pneumonia. Model II included the presence of pneumonia and Heart rate (beats/min) after NIV. Model III included the presence of pneumonia, PaCO_2_ (mmHg) after NIV, and heart rate (beats/min) after NIV. *, *p*-value for the difference in the AUCs compared to Model I, †, *p*-value for comparison between Model II and Model III.

**Table 6 jcm-09-02116-t006:** Changes in physiological parameters after NIV commencement.

Variables	NIV Application (*n* = 78)		*p*-Value
NIV Success (*n* = 46)	Before	After	
Systolic blood pressure(mmHg)	123 (106–142)	117 (104–129)	0.056
Heart rate (beats/min)	92 (82–106)	89 (75–97)	0.007
Respiratory rate (breaths/min)	24 (20–28)	22 (19–25)	0.298
Arterial pH	7.34 (7.29–7.38)	7.37 (7.30–7.43)	<0.001
PaO_2_/FiO_2_ ratio	214 (142–270)	233 (179–271)	0.298
PaCO_2_ (mmHg)	69 (59–83)	63 (51–75)	<0.001
NIV failure (*n* = 32)			
Systolic blood pressure(mmHg)	126 (112–150)	126 (107–145)	0.462
Heart rate (beats/min)	107 (85–117)	97 (89–114)	0.296
Respiratory rate (breaths/min)	27 (23–32)	24 (21–28)	0.159
Arterial pH	7.34 (7.27–7.43)	7.36 (7.31–7.43)	0.082
PaO_2_/FiO_2_ ratio	199 (100–259)	212 (120–280)	0.247
PaCO_2_ (mmHg)	53 (43–68)	48 (39–66)	0.005

Abbreviations: NIV, non-invasive ventilation; PaO_2_, arterial partial pressure of oxygen; FiO_2_, fraction of inspired oxygen; PaCO_2_, arterial partial pressure of carbon dioxide. Data are presented as median (25th percentile–75th percentile). The Wilcoxon signed-rank test was used to determine NIV effect for physiologic parameters and laboratory parameters after NIV application.

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
