# Peer review of "Factors for Predicting Noninvasive Ventilation Failure in Elderly Patients with Respiratory Failure"

_jcm, 2020, doi:10.3390/jcm9072116_

Round 1

Reviewer 1 Report

Dear Editor,

The study “Factors for predicting noninvasive ventilation failure in elderly patients with respiratory failure” touches on an important topic of the role of NIV in elderly patients with acute respiratory failure. The study has no major methodological errors, however, the study does not add too much to the current body of knowledge in this field.

There are a few methodological shortcomings, which should be fixed before acceptance for publication:

  1. Introduction is a little bit to wordy (e.g. sentence : “clinicians should use NIV only when it is indicated”) and not focused on the main topic: factors predicting NIV failure.
  2.  
  3. Methods :
    1. Inclusion/exclusion criteria are not clear”
      1. Authors should define the term ”respiratory failure”, rather than BMI, cardiovascular or cerebrovascular diseases.
      2. Were all eligible subjects admitted due to acute respiratory failure or respiratory failure developed later on during hospital stay?
      3. Criteria to start NIV have to be given.
      4. All contraindication to NIV considered as exclusion criteria should be given, not only examples.
      5. The criteria of NIV failure and indications to intubation, as well as NIV success and weaning should be detailed (e.g. pH>7,35)
    2. The flow chart should be included, it would be very helpful in understanding the enrolment process.
    3. The settings of the treatment is not clear. Were all patients treated in ICU or in HDU and transferred to ICU for invasive ventilation ?
    4.  
  4. Results :
    1. Authors distinguish 3 causes of NIV application: hypercapnic and hypoxemic respiratory failure and cardiogenic pulmonary edema. However, the latter is one of the causes of respiratory failure. All causes of respiratory failure should given (e.g. acute exacerbation of COPD, pneumonia, ARDS, etc..) and analyzed separately from the kind of respiratory failure (namely hypercapnic or hypoxemic)
    2. Such factors as: mean NIV settings, interface type, time of NIV during a day tolerance of NIV, should be included in patients characteristics and considered as potential failure/success factors.
    3. Table 2 shows physiologic effect of NIV after 2 hours (if I understand well), where you can see clinical improvement in both groups: improvement in respiratory rate, pH, PaO2/FiO2 and pCO2 (also statistical). These numbers do not explain the failure of NIV. I would advise to show the results showing real failure of NIV and need for intubation.

Reviewer 2 Report

COMMENTS TO THE AUTHOR(S)

General comments

This present manuscript is a prospective observational multicenter study that aims to explore determinants of NIV failure and timing of intubation delay in old patients with respiratory failure.

Overall, interesting topic, useful to the Respiratory clinicians community.

Statistical analysis is appropriate to confirm the hypothesis.

The discussion sometimes is not focused enough.

The quality of writing is fair, the text is written in acceptable English, even though sometimes it looks a bit unclear; correction of some words, time of the verbs and punctuation typos is recommended.

Some relevant references are missed.

The main limitation of the study is the low number of patients that does not allowed to generalize conclusion, especially in the absence of power analysis.

Major comments

Power analysis for sample size calculation is recommended

Introduction and discussion need to be improved;  please compare this study with others that found predictors of NIV failure in the discussion.

Please specify that the results cannot be generalized due to the low sample size.

Specific comments

  1. Introduction line46-48 please add a reference to this sentence
  2. Introduction line 50- 51 provide a reference for a statement like this
  3. Introduction line 50- 51 provide a reference for a statement like this
  4. Introduction line 53 please add reference for the “ clinical guidelines”
  5. Statistical analysis: please specify descriptive statistics, state how you reported the value (i.e. mean and standard deviation or median and interquartile range)
  6. Table 1 is not clearly readable. I would try to change a bit the layout or the font for subheadings such as location of initial NIV application etc.

I would also move SOFA score next to P/F value since it is a clinical evaluation and information on patient severity status.

  1. Results page Table 2 PO2 and FiO2please change to POFiO2
  2. Figure 1 please specify % of NIV failure on the y axis
  3. Punctuation needs to be revised in all the manuscript.
  4. I would suggest adding this important reference to the reference list:

Scala, R. Challenges on non-invasive ventilation to treat acute respiratory failure in the elderly. BMC Pulm Med 16, 150 (2016). https://doi.org/10.1186/s12890-016-0310-5

Piroddi, I. M. G., Barlascini, C., Esquinas, A., Braido, F., Banfi, P., & Nicolini, A. (2016). Non-invasive mechanical ventilation in elderly patients: A narrative review. Geriatrics & Gerontology International, 17(5), 689–696. doi:10.1111/ggi.12810

Reviewer 3 Report

General comments:

This is an interesting report on NIV use in 78 elderly patients with acute respiratory failure. Like other NIV reports, hypercapneic respiratory failure was associated with NIV success more than hypoxemic respiratory failure. Of note is that pneumonia was an independent risk factor for failure.

Specific comments:

Introduction, line 42: NIV use has been linked to VILI in the presence of vigorous patient efforts.

Introduction, line 51: NIV intolerance is common and associated with poor mask fitting, claustrophobia, noise, anxiety, ineffective settings. These should be mentioned. Indeed, the incidence of NIV intolerance (and the reasons for it) should be reported in the results.

Introduction, line 61: Need a reference for statement that elderly patients are more difficult to wean and require more trachs.

Methods, line 81: Please define acute respiratory failure. It appears that chronic NIV users are included in this. How many? Need to then define acute on chronic respiratory failure and criteria for success in this group.

Methods, line 100: As above, would like know how many NIV failures were primarily NIV intolerance and that patient refused the treatment.

Results, Table 2: The format needs to better separate the success group from the failure group. The label also needs to specify 2 hours as the before/after. It is interesting that the PCO2 decreased to a similar degree in both successes and failures but that the drop in PCO2 appears as an independent predictor for success/failure in Table 3. Maybe the authors could comment on this.

Results, Table 3: As a clinician I do not see strong signals in the physiologic changes that would guide me towards declaring NIV a failure and thus proceeding to intubation. What other criteria were used by the authors to consider NIV a failure and then proceed to intubation?

Round 2

Reviewer 1 Report

The manuscript has been improved. The final decision regarding publication I would left at the discretion of the editor.

However, the scientific soundness is very low. It does not add any new data on the use of NIV. Moreover, there is a high risk of bias due to small group and more importantly due to very questionable methodological assumption that physiological changes assessed once after two hours of NIV may predict the succes or failure in the next few days. What is mostly surprising that reduction in PaCO2, which is the expected effect of NIV, turned out to be a predictive factor of NIV failure. Such a weird result is probably a concequence of the methodological error. Lower level of pCO2 in failure group was most likely due to higher number of patients with hypoxemic respiratory failure, in whom the efficacy of NIV is much lower than in patients with hypercapnic respiratory failure.    

Author Response

upload it as the attached file.

Reviewer 2 Report

The authors addressed all my concerns.

I don't have any further comments. 

I would eventually recommend English proof-reading before publication to improve the language style and improve readability.

Author Response

Thank you for your comment. Revised manuscript was received English editing service to improve language style and improve readability.

Reviewer 3 Report

Review – NIV failures revision 1

General comments:

This revision is an improvement and I continue think these carefully collected/analyzed data on NIV use in elderly patients can be a useful addition to the literature. However, I am confused why the authors focus only on the diagnosis pneumonia as a predictor of NIV success/failure. Is there a rationale for addressing only this diagnosis? Wouldn’t it make more sense to explore the effects of other diagnoses such as acute exacerbations of COPD, CHF, asthma, ARDS etc. as well? Indeed, I could not even find how many patients had pneumonia. This report seems incomplete when only one diagnostic category is selected for inclusion in the analyses.

Specific comments:

Introduction, line 44: Unlike NIV, HFNC does not supply any inspiratory pressure.

Introduction, line 51: add “to avoid intubations” to end of sentence.

Definitions, line 82: replace “or” with “on”

Definitions, line 95: replace “and” with “or”

Table 2: The causes (indications) for NIV include only the broad groupings of hypercapnia, hypoxia, and mixed. As noted above, however, the authors are also focused on the specific diagnosis of pneumonia. If the authors really want to explore specific diagnoses, they should include all of the major diagnostic categories they observed. Alternatively, the authors could eliminate all of the analyses/discussion of pneumonia and just stay with the three main causes (indications) for NIV in this Table. I think the information in this paper is useful enough without being sidetracked by the specific pneumonia diagnosis.

Table 2: Modes should be “pressure support” or “assist control”.

Table 4, line 183: replace “predicting” with “analyzed for effects on NIV success/failure”

Table S1: I think this should be in main body of manuscript

Author Response

Upload it as the attached file

Round 3

Reviewer 2 Report

The authors addressed all the points raised and the manuscript ha significantly improved.

Author Response

Thank you for your comment.

Reviewer 3 Report

The manuscript is much improved. My remaining concern is the authors conclusion that NIV should be applied "without hesitation" to elderly patients with pneumonia. However, their results state that pneumonia is an independent risk factor for NIV failure. I think it would be better to state that NIV is reasonable to try in pneumonia patients but should be done with caution. Both abstract and discussion need to be better worded to match results. 

Author Response

Thank you for your comment.
